# Extensive Therapeutic Drug Monitoring of Colistin in Critically Ill Patients Reveals Undetected Risks

**DOI:** 10.3390/microorganisms8030415

**Published:** 2020-03-15

**Authors:** Stefan Felix Ehrentraut, Stefan Muenster, Stefan Kreyer, Nils Ulrich Theuerkauf, Christian Bode, Folkert Steinhagen, Heidi Ehrentraut, Jens-Christian Schewe, Matthias Weber, Christian Putensen, Thomas Muders

**Affiliations:** 1Department of Anesthesiology and Intensive Care Medicine, University Hospital Bonn, Venusberg-Campus 1, 53127 Bonn, Germany; 2MVZ LaborDiagnostik, Am Rüppurrer Schloss 1, 76199 Karlsruhe, Germany

**Keywords:** colistin, sepsis, colistin methanosulfate, therapeutic drug monitoring, intensive care medicine, renal failure, polymyxin E, Acinetobacter baumannii, carbapenem resistant

## Abstract

(1) Background: With the rise of multi-/pan-drug resistant (MDR/PDR) pathogens, the less utilized antibiotic Colistin has made a comeback. Colistin fell out of favor due to its small therapeutic range and high potential for toxicity. Today, it is used again as a last resort substance in treating MDR/PDR pathogens. Although new guidelines with detailed recommendations for Colistin dosing are available, finding the right dose in critically ill patients with renal failure remains difficult. Here, we evaluate the efficiency of the current guidelines’ recommendations by using high resolution therapeutic drug monitoring of Colistin. (2) Methods: We analyzed plasma levels of Colistin and its prodrug colisthimethate sodium (CMS) in 779 samples, drawn from eight PDR-infected ICU patients, using a HPLC-MS/MS approach. The impact of renal function on proper Colistin target levels was assessed. (3) Results: CMS levels did not correlate with Colistin levels. Over-/Underdosing occurred regardless of renal function and mode of renal replacement therapy. Colistin elimination half-time appeared to be longer than previously reported. (4) Conclusion: Following dose recommendations from the most current guidelines does not necessarily lead to adequate Colistin plasma levels. Use of Colistin without therapeutic drug monitoring might be unsafe and guideline adherence does not warrant efficient target levels in critically ill patients.

## 1. Introduction

Multidrug resistant pathogens (MDR) and extensively drug resistant pathogens (XDR) are posing an imminent threat on the clinicians’ ability to battle infections. According to the U.S. Center of Disease Control and Prevention (CDC), more than 2.8 million new antibiotic resistant infections occur in the United States per year [1]. In particular, the growing rate of pathogens resistant to all of the four main antibiotic classes (Pan-drug resistant, PDR) severely limits available options for treatment [2]. Infections with these pathogens lead to an exceptionally high mortality rate in septic patients [3,4]. With this burden, clinicians are forced to resort to older substances such as polymyxin B or polymyxin E/Colistin [5]. Colistin provides a treatment option for otherwise resistant gram-negative strains, especially *Pseudomonas aeruginosa, Klebsiella species* and *Acinetobacter species* [6]. 

Colistin was isolated from *Bacillus polymyxa* subsp. *colistinus* and consists of a cyclic heptapeptide and a tripeptide side chain acetylated at the N-terminus by a fatty acid moiety [7]. Earlier commercially available Colistin consisted of a mixture of several chemically related subcomponents, including the two major components Colistin A and Colistin B, with a known, concentration-dependent (nephro-, neuro-, oto-) toxicity [5,8]. Therefore, a pharmacologically inactive prodrug Colistin methanosulfate (Colistimethate, CMS) has been derivatized, to reduce associated toxicity. Following parenteral application, CMS is hydrolyzed stepwise to active Colistin. Elimination of CMS occurs primarily renally, with about two thirds of CMS undergoing renal excretion. The other factor for CMS clearance is its conversion to Colistin, through which the available CMS portion decreases. The conversion rate depends both on renal clearance, dosing intervals and nonrenal clearance [9]. In contrast, Colistin elimination mechanisms remain unclear [10]. Only a small portion of Colistin is cleared via the kidney. The majority of the excreted portion undergoes tubular reabsorption and thus remains in the patient. Other pathways for Colistin elimination include proteolytic/enzymatic degradation in various organs rich with proteases and peptidases, with the clear portion remaining elusive [11].

Only few pharmacokinetic data are available because at the time of approval, the analytical methods did not allow thorough and reliable pharmacokinetic testing. Recently, case reports and clinical experience showed a poor predictability of the pharmacokinetic situation under different conditions that affect the distribution volume such as renal replacement therapy or extensive fluid therapy [12,13]. This mandates a close monitoring of both, the prodrug and active, forms of Colistin by Therapeutic Drug Monitoring (TDM) However, TDM for Colistin is currently not routinely available. 

With the resurgence of Colistin during the recent decade, much effort has been put into finding the right dose for treatment, which sufficiently eradicates the pathogens while avoiding toxicity. 

Recently, a new consensus guideline for polymyxin treatment has been published, offering guidance in applying the right dose under various clinical conditions and stages of renal impairment [14]. These recommendations are based on population pharmacokinetic/pharmacodynamics (PK/PD) studies. Without a high resolution TDM, no concise prediction for the individual patient is possible. Therefore, dosing is a fickle topic, especially in ICU patients, where renal function and distribution volume can rapidly change.

Here, we present our findings on TDM of CMS and Colistin in critically ill patients suffering from MDR *Acinetobacter baumannii*, *Klebsiella pneumonia* and *Pseudomonas aeruginosa* infection. Special focus is put on the effect of different dosing regimens in long term CMS treatment for achieving sufficient Colistin plasma levels. This study’s aim is not to provide detailed population-based kinetics, as those have been reported elsewhere [13,15], but rather, to highlight pitfalls encountered in daily clinical routine revealed by high resolution TDM. In contrast to former large PK/PD studies, where only few time points per patient were evaluated, this study allows for a more comprehensive evaluation of CMS and Colistin plasma levels from over 750 individual time-points in eight critically ill patients.

## 2. Materials and Methods 

### 2.1. Patient Samples 

Timeframe of sample collection: April 2013 until October 2014. All patients were treated in the mixed surgical/anesthesiological intensive care unit, University Hospital Bonn, and underwent CMS treatment, before clear dosing recommendations became available.

Eight critically ill patients, some of them under renal replacement therapy (continuous venovenous hemodialysis (CVVHD), intermittent HD (IHD)), infected with gram negative MDR pathogens were treated with CMS. Daily measurements of serum creatinine levels and estimation of glomerular filtration rate using the CKD-EPI Creatinine (eGFR CKD-EPI) equation [16] were used to monitor renal function. Samples were collected multiple times daily prior to/after every dose of CMS for trough/peak level determination of CMS and Colistin. After termination of the Colistin treatment course, samples were derived from the individual patient, to ensure complete and timely removal of Colistin. Blood samples were immediately centrifuged and refrigerated. The plasma was frozen and kept at −20 °C until analysis. After analysis samples were kept at −80 °C for long term storage. 

Microbiology testing was performed by the university’s Department of Microbiology. Samples included regular swab testing for screening purposes (nasal, inguinal, anal, tracheal) and specific testing of blood cultures, wound samples for detection of tissue infection, bronchoalveolar lavage (BAL) fluid and urine. 

### 2.2. Dosing

We aimed for target Colistin serum levels of >2 mg/L in patients suffering from *A. baumannii* infection alone, or >4 mg/L in patients with singular *Pseudomonas* infections or combined infections [14,17]. Dosing was done with or without a loading dose of 6–9 million international units (IU) Colistin and subsequent doses of either 3 × 2 million IU/day, 3 × 3 million IU/day, 3 × 4 million IU or 2 × 4.5 million IU, as indicated. 

Dose adjustments have been made according to the measured plasma levels of Colistin, as soon as those became available. 

### 2.3. Sample Preparation and Detection of CMS and Colistin by HPLC-MS/MS

Only few pharmacokinetic data are available because at the time of drug approval, the analytical methods did not allow thorough and reliable pharmacokinetic testing. So far there is no widely available procedure for therapeutic drug monitoring of Colistin. Here we developed and applied a LC-MS/MS-Method for the monitoring of Colistin levels in patients infected with PDR pathogens. 

### 2.4. Liquid Chromatography Mass Spectrometry 

Calibrators were prepared using stock solutions from certified reference material of the European Directorate for the Quality of Medicines (EDQM) and the respective contents were calculated from the constitution data provided by the EDQM. Calibrators and QC samples were prepared in blank plasma.

A total of 50 μL of plasma samples, calibrators and controls were precipitated with 200 μL of ice cold isopropanol containing 0.5 μg/mL polymyxin B as internal standard and 0.02% TFA. After 10 min at −20 °C, centrifuging 5 min with 20,000 *g* at 4 °C, 100 μL of the supernatant was diluted with 400 μL of the appropriate solvent (water or 30% DMSO in water respectively) and put into the auto sampler at 4 °C. LC-MS/MS system: Dionex UltiMate 3000 XRS autosampler, XRS quaternary UHPLC pump, RS Column compartment. Column used: Aeris Peptide 1.7 μm XB-C18 2.1 × 100 mm (Phenomenex), 40 °C. Mobile phases: Gradient starting with 19% lcms-grade acetonitrile (Baker) and 0.05% TFA (Sigma) in water (Milli-Q) to 40% within 1.6 min at a flow rate of 500 μL/min. Mass spectrometer: TSQ-Vantage triple quad mass spectrometer (Thermo Fisher Scientific, San Jose, CA, USA), HESI-source, positive electrospray ionization mode, settings: ionization voltage 3500 V, vaporizer temperature 500 °C, sheath gas flow 40 (arb), auxiliary gas flow 15 (arb), capillary temperature 200 °C and declustering voltage 10 V. For multiple reaction monitoring (MRM), argon was used as collision gas at a pressure of 1.1 × 10^−3^ Torr. Doubly charged parent ions of Colistins (578.5/100.9, 585.4/100.9) and polymyxin B (602.4/100.9) were used for fragmentation and the product ions and collision energies have been optimized for best intensities. A detailed procedure protocol including description of standardization, sample preparation, sample stability etc., is available upon request.

### 2.5. Statistics

All values are expressed as median ± interquartile range (IQR) or mean ± SD as appropriate. Student’s unpaired *t*-test was used to compare pooled CMS or Colistin peak and trough plasma levels. Repeated measurements two-way ANOVA was used to test for differences between CMS and Colistin plasma levels and changes of plasma drug levels at different time points. Post-hoc analysis was performed using Sidak’s multiple comparisons test. A *p*-value < 0.05 was set as level of significance. Statistical and graphical data handling was performed using GraphPad Prism 8.0 (GraphPad Software, San Diego, CA, USA) and Microsoft Excel (Microsoft, Redmond, WA, USA).

### 2.6. Ethics Approval

Due to the retrospective nature of the analysis and full anonymization of patient data, no individual approval was required and need for informed consent was waived by the local ethics committee (Bonn Medical Faculty Ethics Committee, #027-20; 1 January 2020).

## 3. Results and Discussion

### 3.1. Patient Characteristics

Eight MDR-infected, critically ill patients (four female) were treated with Colistin. Two patients (Patient ID 6 and 7) received two treatment courses of Colistin. Patients were admitted to our ICU at different time points of the respective CMS treatment, leading to varying start points of TDM within the treatment cycle. In five patients TDM was initiated at the start, for the remaining patients TDM started during ongoing CMS treatment as indicated. Plasma samples of 779 individual time points have been analyzed. Median age at admission was 61 (minimum 47, maximum 76, IQR 21) years, median bodyweight 85 (minimum 50, maximum 125, IQR 19) kg, median height 169 (minimum 165, maximum 175, IQR 3) cm and median body mass index 28 (minimum 18, maximum 46, IQR 7) kg/m^2^. The median length of stay in the intensive care unit was 89 (minimum 17, maximum 497 IQR 105) days. Disease severity, indicated by the Sequential Organ Failure Assessment Score (SOFA) [18] was 8 (minimum 1, maximum 15, IQR 6) at ICU admission and 10 (minimum 3, maximum 15, IQR 4) at first day of Colistin treatment. Patients were treated with Colistin for a median of 21 days (minimum 3, maximum 54, IQR 29) days. The specific data for each case are shown in Table 1.

Gram negative MDR pathogens were detected in 391 microbiological samples. They comprised of 317 carbapenem resistant *A. baumannii* (CRAB), 53 PDR *P. aeruginosa* and 21 PDR *K. pneumonia* detections.

The mean CMS trough level was 1.6 (±1.5, *n* = 378), mean CMS peak level was 8.7 (±4.1, *n* = 381). The mean Colistin trough and peak levels were 3.45 (±1.3, *n* = 386) and 3.5 (±1.3, *n* = 383), respectively. CMS and Colistin levels from all patients are depicted in Figure 1A. Injection of CMS induces a significant increase of CMS (peak vs. trough levels, *p* < 0.0001). CMS is than converted to Colistin and/or eliminated. No significant difference between Colistin trough and peak levels was detected. Since Colistin peak levels were not elevated after injection of CMS, the majority of CMS appears to be eliminated prior to conversion. This is in accordance with the suggested CMS kinetics [19]. The Colistin trough and peak levels showed a positive correlation (*r*^2^ = 0.76) (Figure 1B). No relevant correlations were observed between CMS levels and Colistin plasma levels (*r*^2^ = 0.1 for CMS trough levels vs. Colistin trough levels; *r*^2^ = 0.05 for CMS peak levels versus Colistin trough levels) (Figure 1C,D). Based on this, clinicians might estimate the relevant plasma Colistin levels by measuring just the Colistin trough levels. Thus, measuring of Colistin peak levels might be omitted from daily routine. No assumption on Colistin plasma levels should be made based on CMS levels of any kind. 

While CMS is excreted primarily via the kidneys [20], Colistin elimination pathways remain unclear. Previously published elimination half-life of Colistin varies widely and values between 1.6 h and 14.4 h have been reported [13,20,21,22,23]. We were able to calculate elimination half-life for Colistin in 6 out of 8 patients. In our cohort, elimination half-time was greatly extended to a mean of 22.0 h (Figure 2A). The individual half-life kinetics, including details on curve fitting, are provided in the Appendix A (Appendix A
*“Elimination half times of Colistin”*
Appendix A). Individual elimination kinetics followed a first-order kinetic. In one patient (Patient 2), we established CRRT in the midst of the elimination time course. This allowed us to forward project kinetics without CRRT and backwards interpolate kinetics with CRRT. Interestingly, these were nearly identical, indicating, that CRRT does not help to eliminate Colistin (Figure 2B). The amount of available CMS relevantly impacts upon Colistin levels, since CMS acts as a pro-drug. Thus, elimination of CMS is key to reduce Colistin plasma levels whereas CMS accumulation is the source for Colistin build up. CMS elimination can be rapidly affected by changes of renal function and distribution volume. CMS trough levels might indicate potential accumulation of CMS and subsequent accumulation of Colistin. This is of potential relevance when renal CMS elimination might be impaired. Hence, close monitoring of CMS and Colistin trough levels might be crucial.

Previous PK/PD studies sampled with high frequency, but only at a few time points (e.g., once at day 1, day 4, day 6) [15,21,24] and not continuously over the complete treatment course. This approach, in patient cohorts with a high risk of rapidly changing renal function and distribution volumes, and employment of renal replacement therapy, holds the risks of undetected CMS and Colistin plasma level changes.

To the best of our knowledge, our study is the first one providing long-term time course data from 779 time points, allowing for a more detailed evaluation of Colistin plasma levels. The individual data are presented according to the following structure: brief medical history, renal function at start of CMS treatment, used CMS doses, observed CMS/Colistin levels, clinical result of treatment. This is followed by comparison of the findings against current literature and guidelines.

### 3.2. Effects on Acute Renal Insufficiency without CRRT on Achieved Colistin Plasma Levels

Patient 1: At hospital admission, the patient presented with septic shock including acute renal failure. The most likely focus was a gangrenous wound infection of the hand. Colonization with CRAB was known from a previous hospital episode and CRAB infection was confirmed in the necrotic wound. Initially, the patient showed signs of acute septic renal injury with an impaired GFR of ~15 mL/min/1.73 m^2^ (Figure 3A). With renal recovery, no CRRT was necessary (Figure 3A). Colistin treatment was initiated without a loading dose to prevent further kidney injury. A dose of 3 million IU was administered three times per day (Figure 3B), attempting to achieve target levels of >2 mg/L. TDM was initiated at day 3 of CMS treatment. With the selected dose, the patient never achieved persistent Colistin plasma levels within the target range.

In this case, eradication of the CRAB wound infection was achieved. Colonization with CRAB and PDR *Klebsiella pneumonia* persisted. With severe septic renal impairment, the selected dose was too high, at least considering the now available recommendations [14]. Current guidelines would have mandated a daily dose of approximately 4–6 million IU, divided in two doses. Regardless of our seemingly overdosing, the patient never achieved a Colistin concentration at steady state (C_ss,avg_) >2 mg/L, possibly due to the omission of a loading dose and the sustained recovery of renal function. The proposed higher conversion of CMS to Colistin in renal failure with longer half-life [25] did not occur in this specific case. Guideline adherence would have resulted in application of lower doses. In addition, the absence of TDM, would result in undetected underdosing of Colistin. 

### 3.3. Development of Colistin Accumulation and Renal Failure following Increased Colistin Doses, Mandated by Colistin Target Levels 

Patient 2: The patient presented with CRAB wound, CRAB bloodstream and CRAB pulmonary infection, mandating ICU admission. Renal function was unimpaired at admission to our ICU (eGFR 141 mL/min/1.73 m^2^, Figure 4A). CMS treatment was initiated using a loading dose of 6 million IU followed by 3 × 2 million IU daily aiming for Colistin target levels of >2 mg/L. TDM began at day 10 of CMS treatment. 

When the first TDM results were available, insufficient target levels became apparent (Figure 4C). Hence, the dose was increased to 3 × 4 million IU, thus achieving levels >2 mg/L 48 h after dose increase (after a total of 16 days of CMS treatment). For severe wound infections, this is an unacceptably long delay. With the increased dose, we saw a gradual decline of renal function (Figure 4A) with accumulation of CMS and Colistin (gray shaded portion below CMS trough levels, Figure 4A), resulting in termination of CMS applications. Subsequently CRRT became necessary due to sustained renal failure. Colistin treatment cured the bloodstream infection, but CRAB colonization and pulmonary infection remained. Initially, treatment was started with an insufficient dose. Insufficient Colistin plasma levels became apparent through TDM. The prolonged time for achieving target levels >2 mg/L following dose increase was revealed by TDM. Based on current guidelines, a daily dose of 360 mg (~11 million IU) CMS is recommended. In this case we achieved sufficient levels at lower doses (~9 million IU per day, split into three equal doses). The lower dose, compared to the guidelines’ recommendations might explain the delayed response. However, with this reduced dose, renal impairment was observed regardless, indicated by a decreasing eGFR. Hence, an auxiliary dose, in the sense of a secondary “loading dose”, might have alleviated this effect. However, this would most likely have resulted in further renal impairment. Earlier initiation of CRRT might have helped to prevent accumulation of CMS potentially avoiding Colistin accumulation. Honoré and colleagues provide evidence that using CRRT as a “nephro shield” can allow for higher doses of CMS. This allows for up to 4.5 million IU three times per day, when hyperadsorptive filters are used [26,27,28]. Further evidence for this notion is provided by Schmidt et al. [24], who show significant elimination of both CMS and Colistin of up to 50% when prolonged intermittent RRT is employed. That this approach is not necessarily working is shown in some of the following patients.

### 3.4. Effective Colistin Treatment Led to Accumulation Despite of a “Nephro Shield” Strategy with CRRT

Patient 3: In this patient a combination of CRAB bloodstream and wound infection led to admission. Renal function was sustained (Figure 5A, eGFR ~93 mL/min/1.73 m^2^).

Initially, the patient was treated with 3 × 3 million IU CMS (Figure 5B). TDM was started at day 11 of CMS administration. Sufficient Css,avg >2 mg/L were achieved (Figure 5C). The infection persisted, indicated by ongoing positive CRAB cultures from blood samples and wound swabs. Hence, the dose was increased to 3 × 4 million IU (Figure 5B, Day 16 of CMS treatment). This resulted in higher Css,avg levels and subsequent eradication of both bloodstream and wound infection. Additionally, CRAB colonization was eradicated at two of three sites. CMS accumulation was observed towards the end of the treatment course (Figure 5C, days 26–29), despite further improvement of renal function, indicated by increasing eGFR (Figure 5A). CRRT was initiated to prevent further accumulation, following the notion of a “nephro shield” strategy, proposed by Honoré et al. [28]. A further accumulation of Colistin was not detected. No elimination kinetics were available for this patient due to his demise. The used doses both fall within the range of those recommended by the current guidelines. The increased dose might have become necessary due to returning kidney function. However, CMS accumulation was detectable by TDM. Without close monitoring, the CMS accumulation would have failed to be noted, especially with maintained kidney function, putting the patient at risk of subsequent renal injury.

### 3.5. Elimination of CMS trough CRRT can Impair/Prevent Achievement of Sufficient Drug Target Levels

Patient 4: Admitted for gangrenous wound infection of the leg, the patient was treated with stepwise amputation. Renal function was compromised with an initial eGFR of ~24 mL/min/1.73 m^2^ (Figure 6A). Colistin treatment was initiated with a loading dose of 9 million IU, followed by 3 × 3 million IU CMS daily (Figure 6B). With a known infection with both CRAB and MDR *P. aeruginosa*, target Colistin levels in the plasma were set to 4 mg/L, mandating high dose CMS treatment. TDM began at the start of the treatment cycle. The used dose led to sufficient Colistin levels within eight hours of treatment (Figure 6C). 

With further decreasing renal function, dose was lowered to a new dose of 3 × 2 million IU at day 8, corresponding to the recommended dose at this patients’ respective GFR [15]. Regardless, CMS levels continuously increased, providing a constant reservoir of prodrug, maintaining high Colistin plasma levels (Figure 6C). When renal function deteriorated to the point of renal failure, CRRT was started (day 12). With the initiation of continuous renal replacement therapy via CVVHDF, CMS was rapidly eliminated. Colistin plasma levels subsequently decreased below target levels (Figure 6C, day 12 onwards). In order to achieve sufficient drug levels, the dose was increased to 3 × 4 million IU per day (Figure 6B, day 16). With this treatment regimen, in combination with surgical focus eradication, the CRAB infection was eliminated. The pulmonary MDR *Pseudomonas* infection and the MDR *Pseudomonas* colonization persisted.

Despite a reduced CMS dose from day eight onwards, CMS accumulation occurred and high Colistin plasma levels persisted. An intermission of treatment with ongoing monitoring might have been a better approach. Splitting the daily dose into two individual applications, as suggested by others [13,14,29], might not be sufficient to prevent elevated CMS levels. As extended interval dosage can lead to ineffective treatment [30], especially when done without TDM, a reduced maintenance dose following a normal loading dose, as suggested by Schmidt et al. [24] might be better. Insufficient doses potentially favor selection of Colistin resistant species. We observed new occurrence of Colistin resistance in two patients (patients 6&7) with exceptionally long treatment courses, as presented in more detail later.

Initiation of CRRT immediately eliminated CMS and consecutively causing lowered Colistin plasma levels. After dose increase, the target Css,avg of ~4 mg/L was achieved after four days of higher dosing(Figure 6C). To shorten this timeframe, we suggest an auxiliary/secondary loading dose of CMS with 9 million IU when Colistin plasma levels do not promptly rise. This mandates availability of daily TDM because rapid loss of adequate plasma levels and insufficient increase of Colistin plasma levels following dose adjustments would fail to register.

### 3.6. Failure to Rescue: IHD Strategy Does not Necessarily Prevent Accumulation of Potentially Toxic Colistin Drug Levels 

Patient 5: The patients’ hospital admission was originally due to hematooncological disease. While undergoing treatment, pneumonia mandated ICU admission. During this episode bloodstream, soft tissue, pulmonary, urinary, and deep tissue infection with CRAB was detected. The patient was on CRRT prior to the beginning of CMS treatment, with an eGFR of ~75 mL/min/1.73 m^2^ (Figure 7A). The patient was initially treated with 3 × 2 million IU CMS per day (Figure 7B) outside our department. The Colistin plasma target levels were defined as >2 mg/L. TDM was initiated immediately after referral to our department, starting at day 30 of CMS treatment. First results revealed a Colistin Css,avg <2 mg/L (Figure 7C). Hence, the dose was increased to 3 × 4 million IU (Figure 7B). Colistin Css,avg and CMS peak levels increased to sufficient levels (Figure 7C). Despite of CRRT, colistin plasma levels accumulated. This was unchanged by switching to intermittent hemodialysis from day 44 onwards. With daily IHD, plasma Colistin levels remained high (>4 mg/L), while CMS trough levels fell after daily IHD (Figure 7A,C). The CMS treatment successfully eradicated CRAB from the bloodstream and from pulmonary samples. Colonization with CRAB, including wound swabs, persisted. The clinical condition improved to the point of discharge from ICU.

These findings indicate, that use of both, CRRT and IHD, might eliminate the pro-drug CMS. However, this does not necessarily implicate reduced plasma levels of the active component Colistin. The same finding occurred in another patient undergoing IHD from the beginning of Colistin treatment. IHD-adjusted dosing in combination with a loading dose of 9 million IU led to rapid accumulation of both CMS and Colistin, mandating cessation of CMS treatment after 3 days (Patient 8; data not shown due to brevity of time course). These findings underline the importance of TDM with any change of dose adjustment, with or without CRRT, regardless of its mode.

### 3.7. High Colistin Plasma Levels do not Prevent Development of Colistin Resistance, Despite Prolonged Treatment

Patient 6: This patient was admitted to the medical ICU for combined liver and renal failure and underwent CRRT. Suffering from bloodstream and pulmonary CRAB infection alongside with surface colonization, CMS treatment was started without TDM. The observed eGFR during CRRT was ~100 mL/min/1.73 m^2^ (Figure 8A). Following liver transplantation, the patient was transferred to our ICU. At that time, the patient had already been treated with CMS for 28 days. The CMS dose was 3 × 3 million IU at the time of referral to our ICU (Figure 8B). We began TDM and observed levels above the target. The average concentration at steady state was constantly above the recommended 2 mg/L [14] and sometimes exceeded 4 mg/L, the dose normally associated with nephrotoxicity (Figure 8C). Use of continuous renal replacement therapy (CRRT) did not lower the residual CMS levels. Colistin levels were constant throughout the day and unaffected by CRRT (Figure 8D). Under these high Colistin levels, bloodstream infection was cured but colonization persisted. Treatment was discontinued after observing a newly developed Colistin resistance in surface swabs. After 46 days without Colistin, a recurrent bloodstream infection with a Colistin sensible CRAB was observed. Due to the lack of other treatment options a further treatment cycle with CMS was started, despite the still present colonization with Colistin resistant CRAB. This mandated Colistin target levels of >4 mg/l. At this point, due to clinical stabilization, the patient had already been switched from CRRT to intermittent hemodialysis (IHD) at the indicated days (Figure 8E). The observed eGFR at the start of the second treatment cycle was ~66 mL/min/1.73 m^2^ (Figure 8E). A loading dose of 9 million IU was administered, followed by 3 million IU CMS three times a day (Figure 8F). This led to Colistin plasma levels between 3–4 mg/L within 8–16 h (Figure 8G). During this cycle Colistin plasma levels were additionally measured immediately prior to IHD (Figure 8H). IHD (lasting 3–4 h) significantly reduced Colistin plasma levels below 4 mg/L (Figure 8H), mandating higher concentrations at steady state (Figure 8H). With this regime, sufficiently high doses were achieved, and blood stream samples showed eradication of CRAB within 3 days of treatment. Colonization with the Colistin resistant strain persisted.

The current consensus guidelines recommend a daily dose of ~13.3 million IU, in two individual doses. Despite a reduced dose compared to guideline recommendations, we observed high Colistin plasma levels within the critical range during the first treatment cycle. A further dose reduction or cessation/intermission of CMS administrations might have been appropriate. In contrast to patient 4 (Figure 6C) CRRT did not prevent high CMS and Colistin levels in this case. As Honoré and colleagues pointed out themselves, “caution is needed” when using a nephro-shield strategy. Due to multiple factors such as filter properties and anticoagulation, results may vary [26]. During the second treatment cycle, a CMS dose lower than the dose recommend for achieving target levels of ~2 mg/l was used. This in spite of an increased target levels. However, sufficient target levels close to 4 mg/l were observed, even at this lower than recommended dose. Following the guidelines, especially without TDM, this patient might have been at increased risk of inadequate Colistin plasma levels. 

The limited use of Colistin in the past offered some protection against the emergence of bacterial Colistin-resistance, now leading to the recent revival of this substance. With Colistin’s comeback, development of Colistin-resistance in gram-negative strains has been described as a growing phenomenon [31,32,33,34,35,36]. The underlying mechanism of resistance [32] and its origin (induced or selected) in this case can only be speculated upon. As observed here, maintaining high Colistin plasma levels for an extended period did not prevent formation/occurrence of Colistin-resistance.

### 3.8. Constant Colistin Levels can be Achieved with Lowered Doses at Higher Intervals under CRRT. Extended Interval Application might Fail

Patient 7: Initially suffering from wound and pulmonary CRAB infection, the patient’s renal function mandated continuous renal replacement therapy (Figure 9A). With an initial loading, of 9 million IU and a daily dose of 3 million IU three times per day (Figure 9B), the patient promptly achieved a steady state of Colistin concentrations above 2 mg/L (Figure 9C). CMS was discontinued after 6 days after clinical stabilization and successful eradication of CRAB from wound samples. Following a 30-day interval without treatment, Colistin-resistant colonization was detected. Due to a concomitant occurrence of positive blood stream sampling for Colistin-sensible CRAB, another treatment cycle with CMS was initiated. This time with 4.5 million IU twice daily, following a loading dose of 9 million IU. With this extended dose interval, it took three days for concentrations to remain close to or above 2 mg/mL.

Current guidelines recommend a daily dose of ~13.3 million IU divided in two doses. During the first cycle we treated with lower doses, but at shorter intervals. This led to steady state concentrations close to 4 mg/L, despite CRRT. No build-up of CMS was observed with this approach. During the second treatment cycle, the same dose of ~9 million IU per day, at an extended interval (4.5 million IU twice daily), was used. This, with the idea of preventing higher than targeted levels observed during the first treatment cycle. Despite using a loading dose, target Colistin plasma levels were achieved after three days. For a critically ill patient with blood stream infection, this is an arguably long timeframe. However, the bloodstream infection was successfully eradicated. Colonization with the Colistin-resistant PDR strains remained. Using the current guidelines’ recommendations, the patient would have received higher CMS doses with an increased risk of toxicity. Any overly high levels of either CMS or Colistin would have remained undetected without TDM.

In summary, we observed the following pitfalls for CMS treatment and CMS/Colistin monitoring in critically ill patients:Monitoring of renal function on a daily basis is crucial. However, adherence to the guideline-based dose adjustments for renal function/RRT does not necessarily facilitate adequate CMS/Colistin plasma levels. Both, over- and underdosing, were observed.DAILY monitoring of both Colistin and CMS trough levels is necessary, as Colistin plasma levels do not correlate with CMS levels (Figure 1B–D). This might prevent unobserved accumulation of Colistin, if TDM is available in a timely fashion. Monitoring of CMS trough levels allows for detection of pro-drug accumulation.Colistin elimination appears to be unchanged by CRRT and is prolonged in critically ill patients (Figure 2).When Colistin accumulation is observed, dose adjustment can be insufficient. Hence, intermission of Colistin might be more prudent than dose reduction (Figure 5 and Figure 7).Use of CRRT/IHD might rapidly eliminate CMS and thus diminish the available pro-drug when accumulation is observed. CRRT can lead to underdosing, thereby mandating a dose increase (Figure 6). On the other hand, overdosing might still occur/persist despite RRT. Without extensive TDM, both phenomena would remain undetected.Early termination of treatment should be considered, if accumulation occurs, since omission of loading dose can lead to prolonged times with Colistin below target levels (Figure 3, Figure 4 and Figure 6). Thus, use of a loading dose to rapidly achieve relevant target levels, as suggested by others [13,15,24,37], appears to be mandatory. Furthermore, an auxiliary dose might help to rapidly achieve target levels when dose increase becomes necessary.

### 3.9. Study Limitations

Our study has several limitations. First, it is a single center study with a retrospective design. Second, there was no directly measurable/predefined outcome (i.e., ICU mortality), since the patient cohort was small and no direct control group is available. Third, this study cannot provide insights on direct impact on renal failure/nephrotoxicity of Colistin, since no direct causality can be established. Fourth, the patient cohort is small, due to a limited outbreak event. Fifth, no systematic PK/PD analyses were performed due to the small patient cohort. However, we can conclusively demonstrate the inconsistency of Colistin plasma levels despite following the current dosing guidelines. This with a granularity of ~780 individual sampling points, which has so far not been published. 

## 4. Conclusions

In MDR resistant pathogens, Colistin is often the last resort for treatment. Despite the best efforts of accurately describing PK/PD data for Colistin in critically ill patients, adherence to recommendations has a high risk of falling short in achieving the right concentrations to eliminate the infection. This is in part due to the nature of guideline creation based on population kinetics, which has the inherent risk of failing in the individual patient. The created algorithms for CMS treatment regarding renal function lack several important factors, as pointed out by Corona and Cattaneo [38]. Factors such as hypalbuminia, rapid changes of renal function and fast changing distribution volume are often observed in critically ill patients and cannot be disregarded. Our observations underline the necessity of readily available, daily TDM because of the otherwise exceedingly high risk of ineffective/harmful treatment. Without TDM, the assumption of “safe” and “sufficient” doses might be overrated, since both underdosing and accumulation were observed in our cohort. Not using TDM has severe implications for the continuous use of CMS, especially in regard to development of resistance against a last resort drug. We emphasize the importance of using a loading dose as the most vital step of CMS use in the treatment of severe infections, to rapidly achieve relevant CMS and Colistin levels.

## Figures and Tables

**Figure 1 microorganisms-08-00415-f001:**
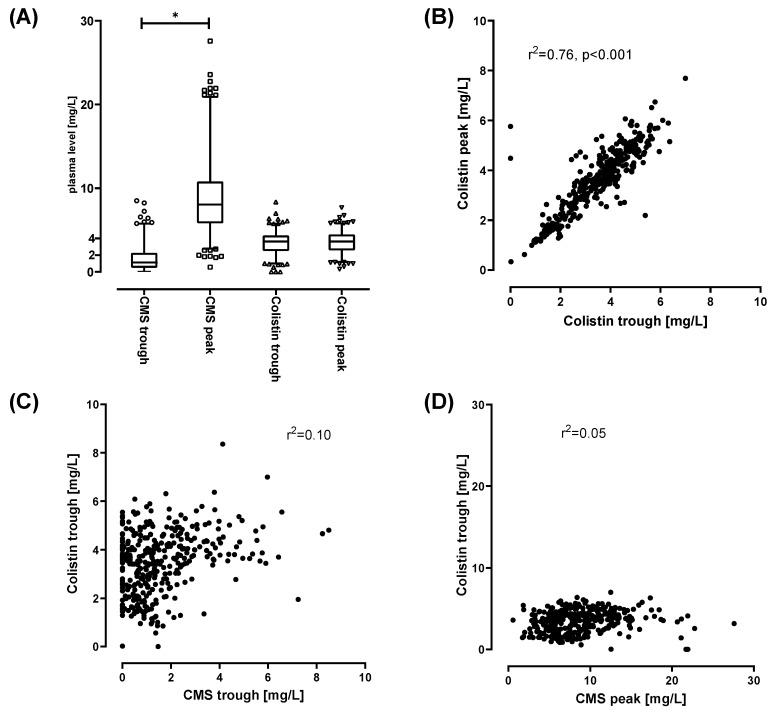
(**A**): range of observed Colistimethate sodium (CMS) and Colistin levels of samples taken 30 min after injection (CMS and Colistin peak levels) or immediately prior to injections of CMS (CMS and Colistin trough levels). A total of 779 samples were analyzed (CMS peak *n* = 385, CMS trough *n* = 388, Colistin peak *n* = 388, Colistin trough *n* = 394). Box indicates 25–75th percentile including median. Whiskers indicate 2.5–97.5 percentile. Outliers are plotted individually. * *p* > 0.001 for *t*-test comparing CMS trough vs CMS peak levels. (**B**–**D**): Linear correlation (Pearson) of Colistin peak and trough levels (**B**) CMS trough and Colistin trough levels (**C**) and CMS peak versus Colistin trough levels (**D**).

**Figure 2 microorganisms-08-00415-f002:**
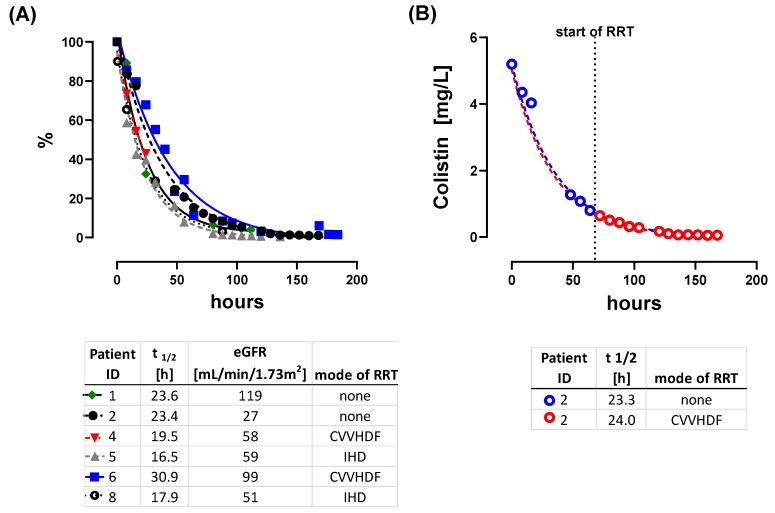
Elimination kinetics of individual patients. Lines show fitted first decay elimination curves. (**A**): normalized elimination kinetics. Patient ID, t_1/2_, estimated glomerular filtration rate (GFR) (eGFR) at the beginning of elimination kinetics and mode of renal replacement therapy (RRT) is indicated. (**B**) Effect of initiation of renal replacement therapy (RRT) on Colistin elimination. After 60 h (dashed line) RRT was started within the elimination period. The blue line is the interpolated elimination kinetic without RRT. Backwards interpolation of samples obtained under RRT (red line) led to an assumed elimination kinetic of Colistin under RRT. CVVHDF = continuous venovenous hemodiafiltration.

**Figure 3 microorganisms-08-00415-f003:**
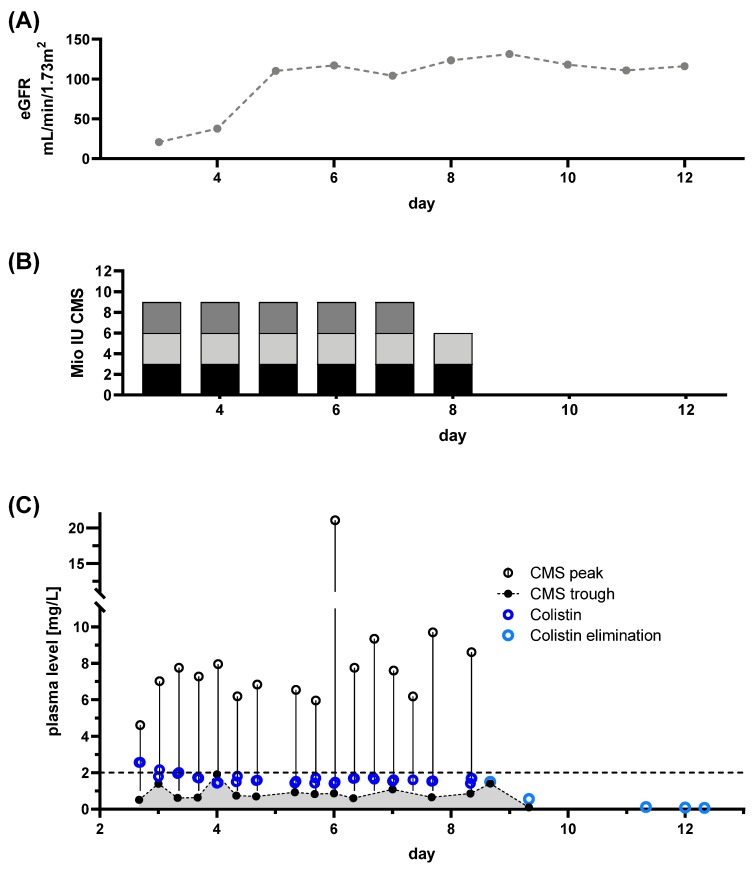
Renal function, dose regimen and drug plasma levels of patient 1: A total of 34 samples were analyzed (colisthimethate sodium (CMS) peak *n* = 16, CMS trough *n* = 16, Colistin peak *n* = 16, Colistin trough *n* = 20). (**A**) eGFR derived from serum creatinine levels is plotted at indicated treatment days. (**B**) Dosing regimen for CMS, colored portions indicate the particular doses during the day. (**C**) Time courses of CMS trough and peak levels along with Colistin levels. Colistin elimination was measured by continuing monitoring after the end of treatment (light blue dots) and the individual fitted elimination is plotted in Figure 1A. Target level for CMS is indicated by dashed line.

**Figure 4 microorganisms-08-00415-f004:**
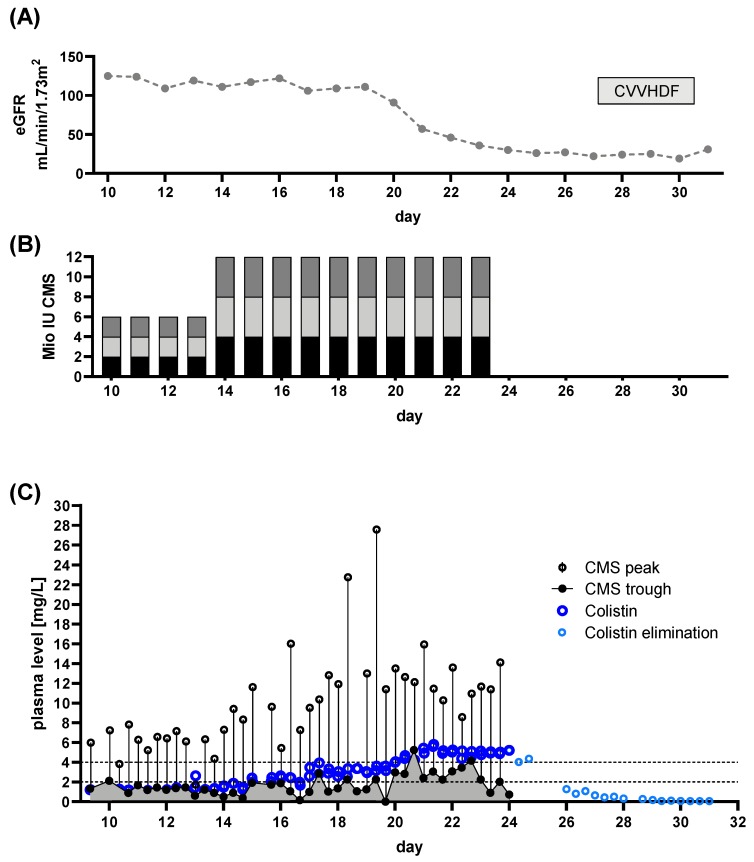
Renal function, dose regimen and drug plasma levels of patient 2: a total of 84 samples were analyzed (colisthimethate sodium (CMS) peak *n* = 42, CMS trough *n* = 42, Colistin peak *n* = 44, Colistin trough *n* = 43). (**A**) estimated glomerular filtration rate (eGFR) derived from serum creatinine levels and renal replacement therapy via continuous venovenous hemodiafiltration (CVVHDF. (**B**) Dose regimen used. Shaded blocks depict the particular individual doses each day. (**C**) Time course of CMS trough and peak levels along with Colistin levels. The CMS reservoir is indicated by the area gray shaded portion of panel (**C**). Colistin elimination was measured by continuing monitoring after the end of treatment (light blue dots) and the individual fitted elimination is plotted in Figure 1A. Target range for Colistin plasma levels is indicated by dashed lines.

**Figure 5 microorganisms-08-00415-f005:**
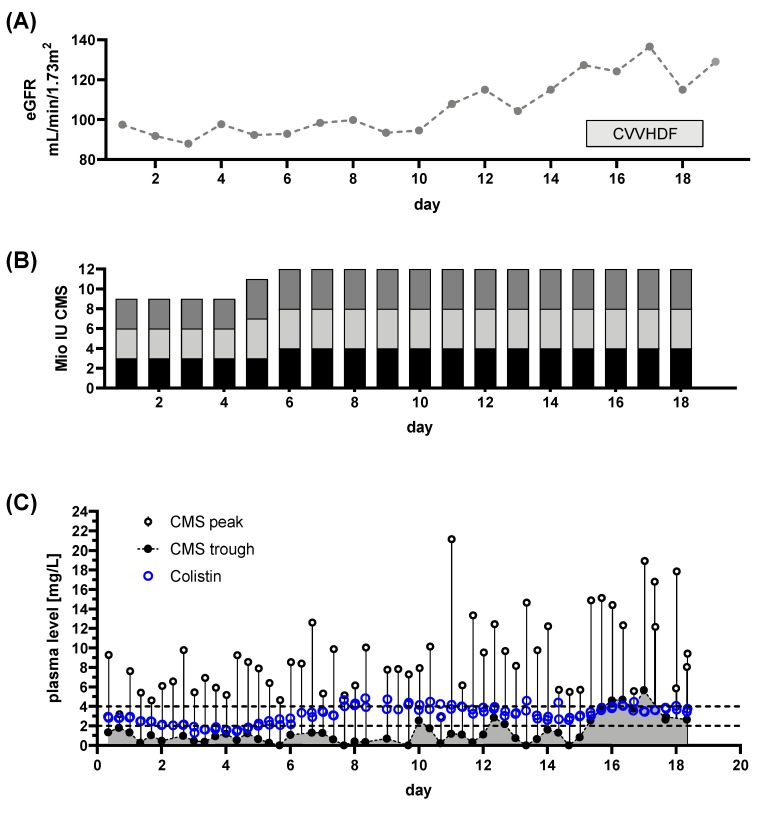
Renal function, dose regimen and drug plasma levels of patient 3: a total of 105 samples were analyzed (colisthimethate sodium (CMS) peak *n* = 56, CMS trough *n* = 50, Colistin peak *n* = 54, Colistin trough *n* = 51). (**A**) estimated glomerular filtration rate (eGFR) derived from serum creatinine levels and renal replacement therapy via continuous venovenous hemodiafiltration (CVVHDF). (**B**) Dose regimen used. Shaded blocks depict the particular individual doses each day. (**C**) Time course of CMS trough and peak levels along with Colistin levels. The CMS reservoir is indicated by the area gray shaded portion of panel (**C**). Target range for Colistin plasma levels is indicated by dashed lines.

**Figure 6 microorganisms-08-00415-f006:**
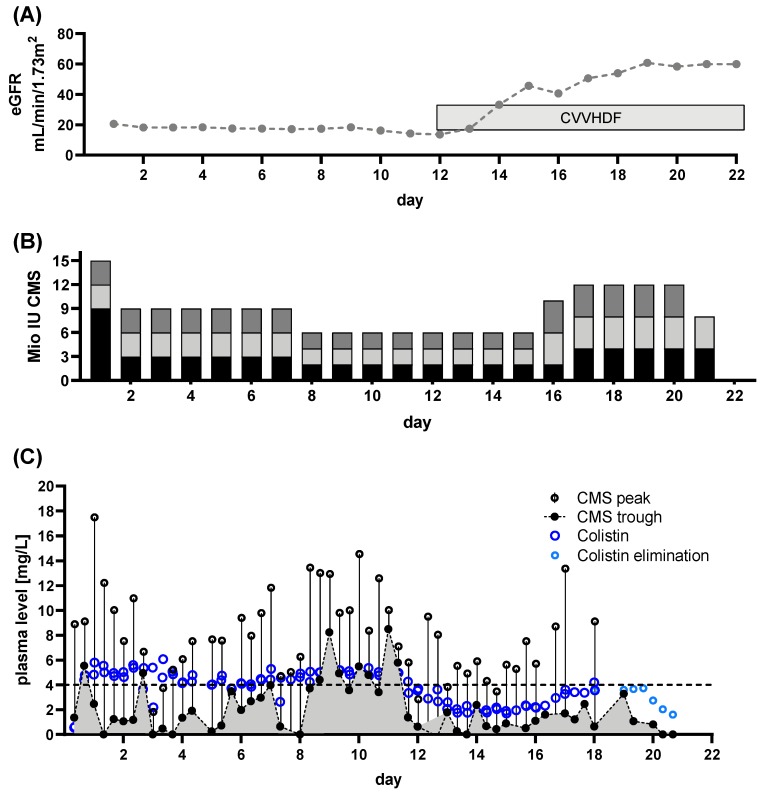
Renal function, dose regimen and drug plasma levels of patient 4: a total of 105 samples were analyzed (colisthimethate sodium (CMS) peak *n* = 51, CMS trough *n* = 54, Colistin peak *n* = 50, Colistin trough *n* = 55). (**A**) estimated glomerular filtration rate (eGFR) derived from serum creatinine levels and renal replacement therapy via continuous venovenous hemodiafiltration (CVVHDF). (**B**) Dose regimen used. Shaded blocks depict the particular individual doses each day. (**C**) Time course of CMS trough and peak levels along with Colistin levels. The CMS reservoir is indicated by the area gray shaded portion of panel (**C**). Colistin elimination was measured by continuing monitoring after the end of treatment (light blue dots) and the individual fitted elimination is plotted in Figure 1A. Target range for Colistin plasma levels is indicated by dashed line.

**Figure 7 microorganisms-08-00415-f007:**
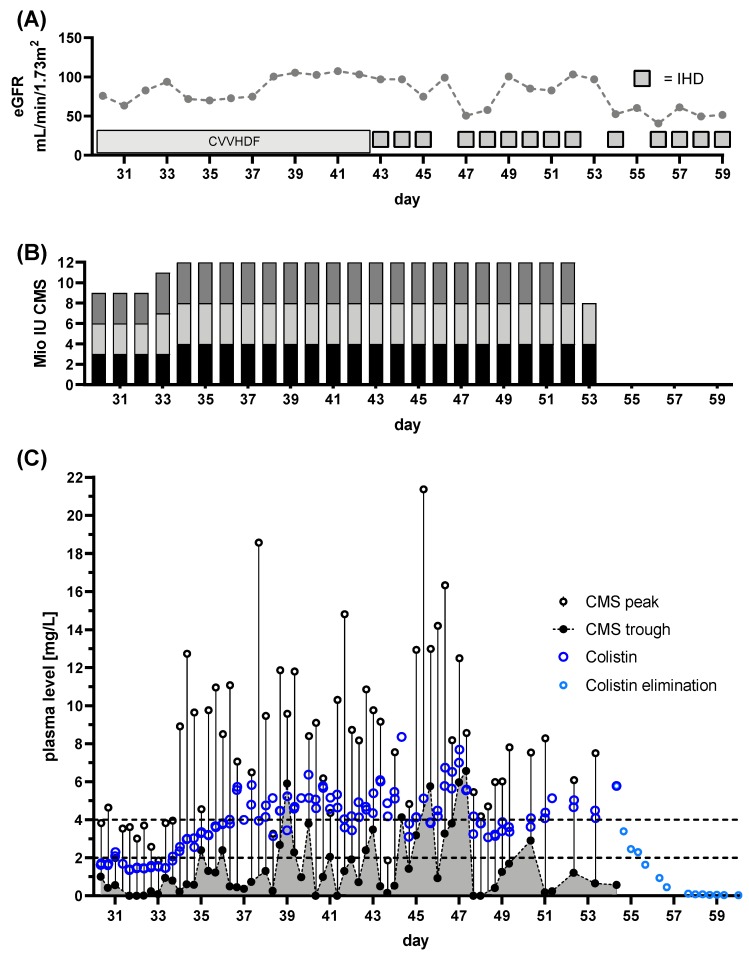
Renal function, dose regimen and drug plasma levels of patient 5: a total of 132 samples were analyzed (colisthimethate sodium (CMS) peak *n* = 60, CMS trough *n* = 60, Colistin peak *n* = 59, Colistin trough *n* = 59). (**A**) estimated glomerular filtration rate (eGFR) derived from serum creatinine levels and renal replacement therapy via continuous venovenous hemodiafiltration (CVVHDF) or intermittent hemodialysis (IHD). (**B**) Dose regimen used. Shaded blocks depict the particular individual doses each day. (**C**) Time course of CMS trough and peak levels along with Colistin levels. The CMS reservoir is indicated by the area gray shaded portion of panel (**C**). Colistin elimination was measured by continuing monitoring after the end of treatment (light blue dots) and the individual fitted elimination is plotted in Figure 1A. Target range for Colistin plasma levels is indicated by dashed lines.

**Figure 8 microorganisms-08-00415-f008:**
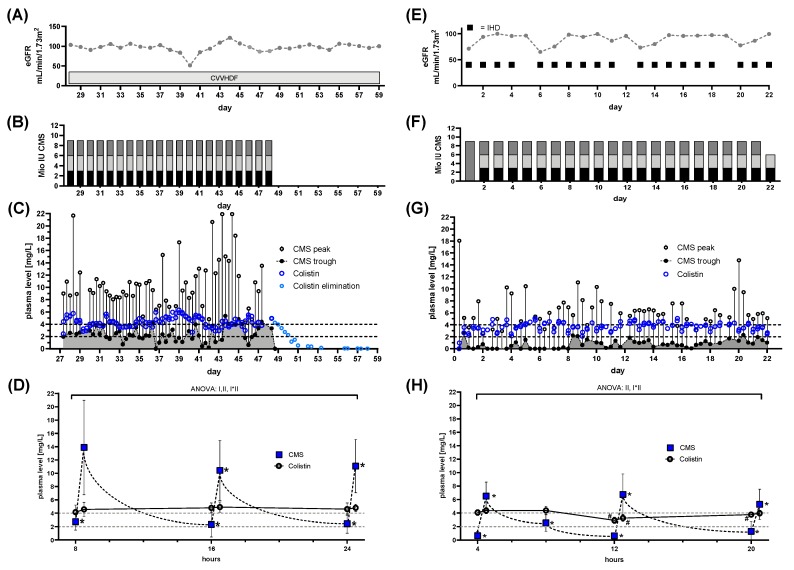
Renal function, dose regimen and drug plasma levels of patient 6: a total of 236 samples were analyzed. First treatment cycle (A)-(D): (colisthimethate sodium (CMS) peak *n* = 59, CMS trough *n* = 55, Colistin peak *n* = 62, Colistin trough *n* = 59). (**A**) estimated glomerular filtration rate (eGFR) derived from serum creatinine levels and renal replacement therapy (RRT) via continuous venovenous hemodiafiltration (CVVHDF). (**B**) Dose regimen used. Shaded blocks depict the particular individual doses each day. (**C**) Time course of CMS trough and peak levels along with Colistin levels. The CMS reservoir is indicated by the area gray shaded portion of panel (**C**). Colistin elimination was measured by continuing monitoring after the end of treatment (light blue dots) and the individual fitted elimination is plotted in Figure 1A. Target range for Colistin plasma levels is indicated by dashed lines. (**D**) Pooled data of daily administrations of CMS or Colistin levels plasma levels during all days with RRT during the first treatment cycle. (**E**–**H**): CMS peak *n* = 56, CMS trough *n* = 59, Colistin peak *n* = 59, Colistin trough *n* = 56. (**A**) eGFR derived from serum creatinine levels and RRT via intermittent hemodialysis (IHD). (**F**) Dose regimen used. Shaded blocks depict the particular individual doses each day. (**G**) Time course of CMS trough and peak levels along with Colistin levels. The CMS reservoir is indicated by the area gray shaded portion of panel (**G**). Target range for Colistin plasma levels is indicated by dashed lines. (**H**) Pooled data of daily administrations of CMS or Colistin levels plasma levels during all days with RRT during the second treatment cycle. Statistics Panel (**D**): mean ± SD; repeated measures ANOVA. Effect I: Differences between CMS and Colistin levels, *p* = 0.0095. Effect II: time effect, *p* < 0.0001. I*II: interaction of effects I&II, *p* < 0.0001. Post-hoc tests: * *p* < 0.05 vs. previous CMS plasma level. Panel (H): mean ± SD; repeated measures ANOVA. Effect I: Differences between CMS and Colistin levels, n.s. Effect II: time effect, *p* < 0.0001. I*II: interaction of effects I&II, *p* < 0.0001. Post-hoc tests: # *p* < 0.05 vs. previous Colistin plasma level, * *p* < 0.05 vs. previous CMS plasma level.

**Figure 9 microorganisms-08-00415-f009:**
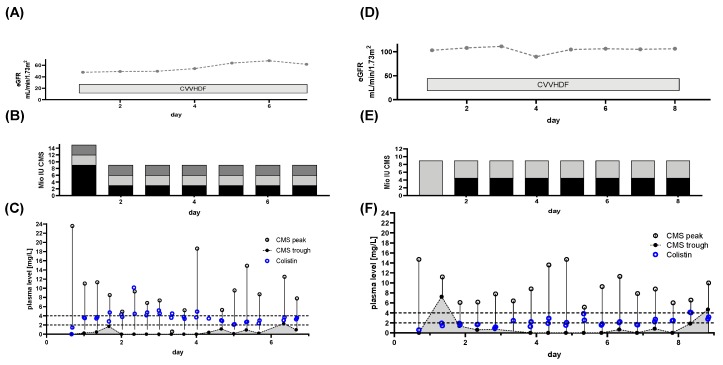
Renal function, dose regimen and drug plasma levels of patient 7 First treatment cycle (**A**–**C**)a total of 35 samples were analyzed (CMS peak *n* = 17, CMS trough *n* = 18, Colistin peak *n* = 17, Colistin trough *n* = 17). During the 2nd treatment cycle (**D**–**F**) a total of 32 samples, CMS peak *n* = 17, CMS trough *n* = 16, Colistin peak *n*= 17, Colistin trough *n* = 15) were analyzed. (**A**)/(**D**) estimated glomerular filtration rate (eGFR) derived from serum creatinine levels and renal replacement therapy (RRT) via continuous venovenous hemodiafiltration (CVVHDF). (**B**)/(**E**) Dose regimen used. Shaded blocks depict the particular individual doses each day. (**C**)/(**F**) Time course of CMS trough and peak levels along with Colistin levels. The CMS reservoir is indicated by the area gray shaded portion of panel (**C**)/(**F**). Target range for Colistin plasma levels is indicated by dashed lines.

**Table 1 microorganisms-08-00415-t001:** Epidemiology and case specific details regarding infection, disease severity and need for renal replacement therapy.

Pat ID	Age at Admission [y]	Weight [kg]	Height [cm]	BMI [kg/m^2^]	ICU LOS [d]	Number of Days on Colistin	SOFA Score at Admission	SOFA Score at Start of CMS Treatment	Site of Infection	Gram Negative PDR Pathogens	RRT
1	62	85	175	27.8	17	7	10	10	Wound infection, pulmonary infection + colonization	*A. baumannii*, *E. cloacae*, *K. pneumoniae*	None
2	73	85	175	27.8	87	52	7	3	Blood stream, pulmonary+ colonization	*A. baumannii*	Post treatment (CVVHDF)
3	75	100	170	34.0	50	29	7	10	Blood stream, wound infection + colonization	*A. baumannii*, *P. aeruginosa*	CVVHDF
4	76	85	170	29.0	497	19	8	8	Wound infection, pulmonary infection + colonization	*P. aeruginosa*	CVVHDF
5	54	100	168	34.6	145	54	15	15	Blood stream, pulmonary, wound infection, urinary tract infection + colonization	*A. baumannii*	CVVHDF, IHD
6.1	61	50	168	17.7	161	38	14	8	Blood stream, pulmonary+ colonization	*A. baumannii*	CVVHDF
6.2						22	14	11	Blood stream, pulmonary+ colonization	*A. baumannii*	IHD
7.1	49	125	165	45.9	91	6	1	15	Wound infection, pulmonary infection + colonization	*A. baumannii*	CVVHDF
7.2						7	1	9	Blood stream, wound infection (colistin resistant), + colonization	*A. baumannii*	CVVHDF
8	47	68	168	24.1	26	3	7	13	Pulmonary infection, wound infection	*P. aeruginosa*	IHD

BMI: body mass index, LOS: length of stay, SOFA: Sequential Organ Failure Assessment, CMS: Colisthimethate sodium, PDR: pan-drug resistant, RRT: renal replacement therapy, Colonization indicates positive testing in routine surface swabs from the oropharynx, groin, anal sphincter. Pulmonary infection relates to positive testing in tracheal fluid or bronchioalveolar fluid. Wound infection is regarded as positive testing from deep tissue/wounds, excluding surface swabs or superficial wounds. IHD = intermittent hemodialysis, CVVHDF = continuous venovenous hemodiafiltration.

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
