# Peer review of "Extensive Therapeutic Drug Monitoring of Colistin in Critically Ill Patients Reveals Undetected Risks"

_microorganisms, 2020, doi:10.3390/microorganisms8030415_

Round 1
Reviewer 1 Report
This manuscript describes the intensive therapeutic drug monitoring of colistin and its pro-drug colistin methanosulphate in eight patients receiving colistin in critical care.
It is an unusual manuscript given the detailed monitoring conducted, the fact colistin and CMS were both monitored and that each of the eight clinical cases are considered individually and in some detail.
The data given is interesting and worth publication but the format is difficult.
I would suggest the authors reduce the length of the case histories and move some of the text into the discussion. At present the results contain a mix of results and discussion.
There are several themes from the case histories - perhaps these could be combined in the discussion?
In conclusion, the data presented here is of interest but needs significant presentational changes with shortening of the manuscript and a clear focus.
Author Response
Reviewer 1
Comments and Suggestions for Authors
- This manuscript describes the intensive therapeutic drug monitoring of colistin and its pro-drug colistin methanosulphate in eight patients receiving colistin in critical care.
- It is an unusual manuscript given the detailed monitoring conducted, the fact colistin and CMS were both monitored and that each of the eight clinical cases are considered individually and in some detail.
- The data given is interesting and worth publication but the format is difficult.
- I would suggest the authors reduce the length of the case histories and move some of the text into the discussion. At present the results contain a mix of results and discussion.
Reply: We thank the reviewer for his/her comment. In regard to the structure of the Results & Discussion section we chose to follow the journals option to do a joint section (Copied from the “Instructions for Authors section”: Discussion: Authors should discuss the results and how they can be interpreted in perspective of previous studies and of the working hypotheses. […] This section may be combined with Results.)
The individual cases are presented with the same structure for each individual case: brief medical history; Renal function; Used CMS dose and target values; observed Colistin/CMS levels. Clinical result; Discussion of our observations vs published literature and guidelines.
This has been emphasized by incorporating the following section prior to the patient time course data: “The individual data are presented according to the following structure: brief medical history, renal function at start of CMS treatment, used CMS doses, observed CMS/Colistin levels, clinical result of treatment. This is followed by comparison of the findings against current literature and guidelines.” (p9. Lines 215-218) - There are several themes from the case histories - perhaps these could be combined in the discussion?
Reply: As the reviewer points out, there are several themes in the case histories, which follow the changes in Colistin/CMS levels depending on altering renal function. Patient 1 with unaltered/improving renal function to Patient 8 with impaired/replaced renal function over the whole time course of CMS treatment. The conclusions drawn from our observations are already summarized at the end of the combined Results & Discussion section. - In conclusion, the data presented here is of interest but needs significant presentational changes with shortening of the manuscript and a clear focus.
Reply: We thank the reviewer for his/her opinion. However, we believe to have structured the data consistently throughout the manuscript, staying within the journal’s instructions. Changing the presentation at this point would mandate significant changes within the manuscript. Since the second reviewer made no mention of any kind in this direction, we would like to leave this decision to the editor. Regarding manuscript length, we believe that shortening the manuscript would further increase the chance of losing relevant observations from individual cases. As the second reviewer asked for some additional comments within the text regarding CMS/Colistin metabolism, the overall length had to be slightly increased. Nowadays, word count / manuscript length is not limited due to the digital/online nature of journals and the journal gives no limit. Hence, for this comment we’d also value the editor’s opinion. If the editor deems further shortening necessary, we will of course attempt to provide this.
We'd also like to take this opportunity to thank the reviewer for her/his time and effort to review and comment on our work.
Reviewer 2 Report
A reviewer has but to commend the authors for their effort to apply a TDM strategy for colistin therapy. It is indeed a strenuous undertaking, full of methodological issues. Despite their criticism over the recent recommendations for the optimal use of colistin, they embrace one such suggestion, which is the TDM of colistin.
Points to mention:
- The authors fail to comment on the erratic metabolism of the prodrug to active colistin, which may, in part, explain that the levels of the drug either fail short or are overshot.
- They have to emphasize on the importance of loading dose as the most vital step of colistin use in the treatment of severe infections.
Author Response
Reviewer 2
Comments and Suggestions for Authors
A reviewer has but to commend the authors for their effort to apply a TDM strategy for colistin therapy. It is indeed a strenuous undertaking, full of methodological issues. Despite their criticism over the recent recommendations for the optimal use of colistin, they embrace one such suggestion, which is the TDM of colistin.
Points to mention:
- The authors fail to comment on the erratic metabolism of the prodrug to active colistin, which may, in part, explain that the levels of the drug either fail short or are overshot.
Reply: The section on CMS/Colistin conversion and metabolic pathways for CMS/Colistin elimination has been updated in the manuscript. The following passages were added/changed: “Elimination of CMS occurs primarily renaly, with about two thirds of CMS undergoing renal excretion. The other factor for CMS clearance is its conversion to Colistin, through which the available CMS portion decreases. The conversion rate depends both on renal clearance, dosing intervals and non renal clearance [9]. […] Only a small portion of Colistin is cleared via the kidney. The majority of the excreted portion is undergoing tubuluar reabsorption and thus remains in the patient. Other pathways for Colistin elimination include proteolytic/enzymatic degradation in various organs rich with proteases and peptidases, with the clear portion remaining elusive [11].” (page 2, lines 47-54) - They have to emphasize on the importance of loading dose as the most vital step of colistin use in the treatment of severe infections.
Reply: We thank the reviewer for his/her comments and share his view. We incorporated the well versed line into the conclusion.
“We emphasize the importance of using a loading dose as the most vital step of CMS use in the treatment of severe infections, to rapidly achieve relevant CMS and Colistin levels.” (page 21, lines 537-538)
The underlying rationale for this was already included in the brief summary, along with the supporting literature “Omission of loading dose can lead to prolonged times with Colistin below target levels (Figure 3, 4, 6). Thus, use of a loading dose to rapidly achieve relevant target levels, as suggested by others [11,13,22,35], appears to be mandatory. Furthermore, an auxiliary dose might help to rapidly achieve target levels when dose increase becomes necessary.” (page 21, lines 509-511)
We'd also like to take this opportunity to thank the reviewer for her/his time and effort to review and comment on our work.